# KARMA: Leveraging Multi-Agent LLMs for Automated Knowledge Graph Enrichment

**Yuxing Lu[1,2], Wei Wu[1], Xukai Zhao[3], Rui Peng[1], Jinzhuo Wang[1†]**

[1] Department of Big Data and Biomedical AI, Peking University, Beijing, China
[2] Wallace H. Coulter Department of Biomedical Engineering, Georgia Institute of Technology, Atlanta, USA
[3] School of Architecture, Tsinghua University, Beijing, China

† Corresponding author: `wangjinzhuo@pku.edu.cn`

## Abstract

Maintaining comprehensive and up-to-date knowledge graphs (KGs) is critical for modern AI systems, but manual curation struggles to scale with the rapid growth of scientific literature. This paper presents KARMA, a novel framework employing multi-agent large language models (LLMs) to automate KG enrichment through structured analysis of unstructured text. Our approach employs nine collaborative agents, spanning entity discovery, relation extraction, schema alignment, and conflict resolution that iteratively parse documents, verify extracted knowledge, and integrate it into existing graph structures while adhering to domain-specific schema. Experiments on 1,200 PubMed articles from three different domains demonstrate the effectiveness of KARMA in knowledge graph enrichment, with the identification of up to 38,230 new entities while achieving 83.1% LLM-verified correctness and reducing conflict edges by 18.6% through multi-layer assessments.

## 1 Introduction

Knowledge graphs (KGs) are essential for structuring and reasoning over complex information across diverse fields [9, 11, 18]. By encoding entities and their relationships in machine-readable formats, widely adopted KGs such as Wikidata [26] and DBpedia [13] have become foundational to both industry and academic research. Yet, the exponential growth of scientific literature, with over 7 million articles published annually [2], exposes a significant bottleneck: the widening gap between unstructured knowledge in texts and its structured representation in KGs.

The challenge of enriching KGs becomes even more apparent in fields with complex and specialized terminology, such as healthcare, finance, or autonomous systems. Traditional approaches to KG enrichment, such as manual curation, are reliable but unsustainable at scale. Automated methods based on conventional natural language processing (NLP) techniques often struggle to handle domain-specific terminology and context-dependent relationships found in scientific and technical texts [22]. Moreover, extracting and integrating knowledge into existing KGs requires robust mechanisms for schema alignment, consistency, and conflict resolution [5]. In high-stakes applications, the costs of inaccuracies in these systems can be severe.

Recent advances in large language models (LLMs) [7, 1, 15] have demonstrated remarkable improvements in contextual understanding and reasoning [28]. Building on these advances, the research community has increasingly explored multi-agent systems, where several specialized agents work in concert to tackle complex tasks [8]. These systems harness the strengths of individual agents, each optimized for a particular subtask, and enable cross-agent verification and iterative refinement of outputs. Such multi-agent frameworks have shown promise in areas ranging from decision-making to structured data extraction [6, 19], offering robustness through redundancy and collaboration. However,

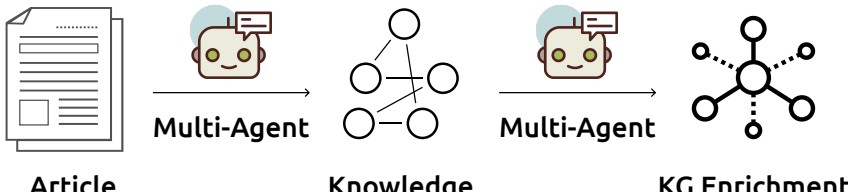

Figure 1: Multi-agent LLM can parse articles into new knowledge, and integrate to existing knowledge graphs through filtering.

directly applying these systems to KG enrichment remains challenging due to issues like domain adaptation, systematic verification requirements [10], and the complexity of integrating outputs into heterogeneous knowledge structures.

In this paper, we propose KARMA, a novel multi-agent framework that harnesses LLMs through a collaborative system of specialized agents (Figure 1). Each agent focuses on distinct tasks in the KG enrichment pipeline. Our framework offers three key innovations. First, the multi-agent architecture enables cross-agent verification, enhancing the reliability of extracted knowledge. For instance, *Relationship Extraction Agents* validate candidate entities against *Schema Alignment* outputs, while *Conflict Resolution Agents* resolve contradictions through LLM-based debate mechanisms. Second, domain-adaptive prompting strategies allow the system to handle specialized contexts while preserving accuracy. Third, the modular design ensures extensibility and supports dynamic updates as new entities or relationships emerge. Through proof-of-concept experiments on datasets from three distinct domains, we demonstrate that KARMA can efficiently extract high-quality knowledge from unstructured texts, substantially enriching existing knowledge graphs with both precision and scalability.

## 2 Related Work

### 2.1 Knowledge Graph Construction

The quest to transform unstructured text into structured knowledge has evolved through three generations of technical paradigms. *First-generation systems (1990s-2010s)* like WordNet [21] and ConceptNet [16] relied on hand-crafted rules and shallow linguistic patterns, achieving high precision at the cost of limited recall and domain specificity. *The neural revolution (2010s-2022)* introduced learned representations through architectures like BioBERT [12] and SapBERT [17], which achieved improvements on biomedical NER through domain-adaptive pretraining. However, these methods require expensive supervised tuning (3-5k labeled examples per relation type [30]) and fail to generalize beyond predefined schema, which is a critical limitation when processing novel scientific discoveries. The *current LLM-powered generation (2022-present)* attempts to overcome schema rigidity through instruction tuning [24, 31]. This progression reveals an unresolved tension: neural methods scale better than rules but require supervision, while LLMs enable open schema learning at the cost of verification mechanisms. LLMs have shown promise in open-domain KG construction through their inherent reasoning capabilities. However, these approaches exhibit critical limitations: (1) Hallucination during extracting complex relationships [20], (2) Inability to maintain schema consistency across documents [29], and (3) Quadratic computational costs when processing full-text articles [23].

### 2.2 Multi-Agent Systems

Early multi-agent systems focused on distributing subtasks across specialized modules, such as separate agents for named entity recognition and relation extraction [3]. These systems relied on predefined pipelines and handcrafted coordination rules, limiting adaptability to new domains. Recent advances in LLMs have enabled more dynamic architectures and rediscovered multi-agent collaboration as a mechanism for enhancing LLM reliability [25, 19]. Building on classic blackboard architectures, contemporary systems like AutoGen [28] show that task decomposition with specialized agents reduces hallucination compared to monolithic models. For knowledge graph construction, [14] demonstrated that task decomposition across specialized agents (e.g., entity linker, relation validator)

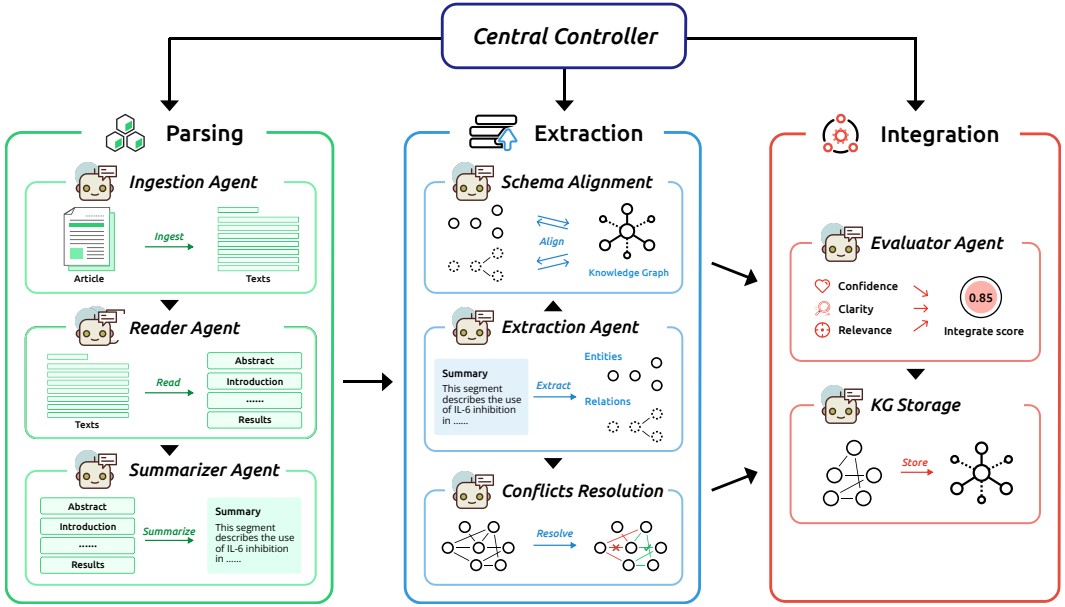

Figure 2: System overview of the KARMA multi-agent architecture. Each agent is an LLM-driven module tasked with specific roles such as ingestion, summarization, entity recognition, relationship extraction, conflict resolution, and final evaluation.

improves schema alignment on Wikidata benchmarks. maintaining linear time complexity relative to input text length.

KARMA synthesizes insights from these research threads while introducing key innovations: (1) a modular, multi-agent architecture that allows for specialized handling of complex tasks in knowledge graph enrichment, (2) domain-adaptive prompting strategies that enable more accurate extraction across diverse scientific fields, (3) LLM-based verification mechanisms that mitigate issues such as hallucination and schema inconsistency.

## 3 Methodology

In this section, we introduce KARMA, a hierarchical multi-agent system (see Figure 2) that leverages specialized LLMs to perform end-to-end KG enrichment. Our approach decomposes the overall task into modular sub-tasks, ranging from document ingestion to final KG integration, each handled by an independent LLM-based agent. We first present a formal problem formulation and then detail the design and mathematical foundations of each agent within the pipeline.

### 3.1 Problem Formulation

Let $\mathcal{G} = (V, E)$ denote an existing KG, where $V$ is the set of entities (e.g., genes, diseases, drugs) and $E$ the set of directed edges representing relationships. Each relationship is defined as a triplet $t = (e_h, r, e_t)$ with $e_h, e_t \in V$ and $r$ specifying the relation type (e.g., treats, causes). We are provided with a corpus of unstructured publications $\mathcal{P} = p_1, \ldots, p_n$. The objective is to automatically extract novel triplets $t \notin E$ from each document $p_i$ and integrate them into $\mathcal{G}$ to form an augmented graph $\mathcal{G}_{\text{new}}$.

$$\mathcal{G}_{\text{new}} = \mathcal{G} \cup \bigcup_{i=1}^{n} \mathcal{K}_i, \text{where } \mathcal{K}_i = \text{Extract}(p_i), \tag{1}$$

where $\text{Extract}(p_i)$ is the set of valid triplets obtained from publication $p_i$. To maintain consistency and accuracy, each candidate triplet is evaluated by an LLM-based verifier prior to integration.

## 3.2 System Overview

KARMA comprises multiple LLM-based agents operating in parallel under the orchestration of a Central Controller. Each agent uses specialized prompts, hyper-parameters, and domain knowledge to optimize its performance. In KARMA, we define a set of agents (B):

- **Ingestion Agents (IA)**: Retrieve and normalize input documents (B.3).
- **Reader Agents (RA)**: Parse and segment relevant text sections (B.4).
- **Summarizer Agents (SA)**: Condense relevant sections into shorter domain-specific summaries (B.5).
- **Entity Extraction Agents (EEA)**: Identify and normalize topic-related entities (B.6).
- **Relationship Extraction Agents (REA)**: Infer relationships between entities (B.7).
- **Schema Alignment Agents (SAA)**: Align entities and relations to KG schemas (B.8).
- **Conflict Resolution Agents (CRA)**: Detect and resolve logical inconsistencies with existing knowledge (B.9).
- **Evaluator Agents (EA)**: Aggregate multiple verification signals and decide on final integration (B.10,B.11,B.12).

### 3.3 Ingestion Agents (IA)

The Ingestion Agents are LLM-based modules specialized in document retrieval, format normalization, and metadata extraction. Let $p_i$ be a raw publication. IA includes:

$$\mathrm{IA}(p_i) \; = \; \Big(\mathrm{normalize}(p_i), \; \mathrm{metadata}(p_i)\Big), \tag{2}$$

where $\mathrm{normalize}(p_i)$ uses an LLM prompt $P_{\mathrm{ingest}}$ to handle complexities like OCR errors, or structural inconsistencies. The output is a standardized textual representation plus key metadata (*journal*, *date*, *authors*, etc.). This representation is then placed into a data queue for *Reader Agents*.

### 3.4 Reader Agents (RA)

Reader Agents parse normalized text into coherent segments (abstract, methods, results, ect.) and filter out irrelevant content. Let $p_i'$ be the normalized document. RA splits $p_i'$ into $\{s_1, s_2, \ldots, s_{m_i}\}$. Each segment $s_j$ is assigned a relevance score $R(s_j)$ by:

$$R(s_j) \; = \; \mathrm{LLM}_{\mathrm{reader}}\big(s_j, \mathcal{G}\big), \tag{3}$$

where $\mathrm{LLM}_{\mathrm{reader}}$ is prompted with domain-specific instructions to assess the segment's biomedical significance relative to the current KG $\mathcal{G}$. RA discards segments if $R(s_j) < \delta$, where $\delta$ is a domain-calibrated threshold. Surviving segments are passed along to Summarizer Agents.

### 3.5 Summarizer Agents (SA)

To reduce computational overhead, each RA segment $s_j$ is condensed by Summarizer Agents into a concise representation $u_j$. Formally, we define:

$$u_j = \mathrm{LLM}_{\mathrm{summ}}\big(s_j, P_{\mathrm{summ}}\big), \tag{4}$$

where $P_{\mathrm{summ}}$ is a prompt for LLM to retain critical entities, relations, and domain-specific terms. This summarization ensures *Entity Extraction Agents* and *Relationship Extraction Agents* receive textual inputs that are both high-signal and low-noise.

### 3.6 Entity Extraction Agents (EEA)

**LLM-Based NER.** Each summary $u_j$ is routed to an LLM-based NER pipeline that identifies mentions of topic-related entities. Define:

$$E(u_j) = \mathrm{LLM}_E\big(u_j, P_E\big) \; \odot \; D_E, \tag{5}$$

where $\mathrm{LLM}_E$ is an specialized entity-extraction LLM with prompt $P_E$, and $\odot D_E$ indicates a dictionary/ontology-based filtering. This step filters out false positives and normalizes entity mentions to canonical forms (e.g., mapping "acetylsalicylic acid" to "Aspirin").

**Entity Normalization.** Let $e$ be a raw entity mentioned from $E(u_j)$. We map $e$ to a normalized entity $\hat{e} \in V$ by minimizing a distance function in a joint embedding space:

$$\hat{e} = \arg\min_{v \in V} \; d\big(\phi(e), \psi(v)\big), \tag{6}$$

where $\phi$ maps textual mentions to embeddings (using, e.g., a BERT-based model), and $\psi$ maps known KG entities to the same embedding space. The distance metric $d(\cdot, \cdot)$ can be cosine distance or a domain-specific measure. Any entity with $\min_{v \in V} d(\phi(e), \psi(v)) > \rho$ is flagged as new and added to the set of candidate vertices $V^+$.

### 3.7 Relationship Extraction Agents (REA)

After entity normalization, each pair $(\hat{e}_i, \hat{e}_j)$ within summary $u_j$ is fed to an LLM-based classifier:

$$p(r \mid \hat{e}_i, \hat{e}_j, u_j) = \text{LLM}_R\big(\hat{e}_i, \hat{e}_j, u_j, P_R\big), \tag{7}$$

where $p(r|\cdot)$ is the probability distribution over possible relationships $r \in \{r_1, \ldots, r_K\}$. The prompt $P_R$ instructs the LLM to focus on domain relationship candidates. We select any relationship $r$ for which $p(r|\hat{e}_i, \hat{e}_j) \geq \theta_r$ and form a triplet $(\hat{e}_i, r, \hat{e}_j)$. In certain passages, more than one relationship can be implied. We allow multi-label predictions by setting an indicator variable:

$$I(r) = \mathbb{I}\{p(r \mid \hat{e}_i, \hat{e}_j) \geq \theta_r\}, \tag{8}$$

Hence, $\mathcal{R}(u_j)$ is the set of triplets $(\hat{e}_i, r, \hat{e}_j)$ such that $I(r) = 1$.

### 3.8 Schema Alignment Agents (SAA)

If a new entity $v \in V^+$ or a new relation $r$ does not match existing KG types, the Schema Alignment Agent performs a domain-specific classification. For entities, the SAA solves:

$$\tau^* = \arg\max_{\tau \in \mathcal{T}} \; \text{LLM}_{\text{SAA}}\big(v, \tau, P_{\text{align}}\big), \tag{9}$$

where $\mathcal{T}$ is the set of valid entity types (`Disease`, `Drug`, `Gene`, etc.), and $\text{LLM}_{\text{SAA}}$ estimates the probability that $v$ belongs to type $\tau$. A similar approach is used for mapping new relation $r$ to known KG relation types. If no suitable match exists, the SAA flags $v$ or $r$ as candidate additions for review.

### 3.9 Conflict Resolution Agents (CRA)

New triplets can contradict previously established relationships. Let $t = (\hat{e}_h, r, \hat{e}_t)$ be a newly extracted triplet, and let $t' = (\hat{e}_h, r', \hat{e}_t)$ be a conflicting triplet in $\mathcal{G}$ if $r$ is logically incompatible with $r'$. We define:

$$\text{conflict}(t, \mathcal{G}) = \begin{cases} 1, & \text{if } \exists\, t' \text{ that contradicts } t, \\ 0, & \text{otherwise.} \end{cases} \tag{10}$$

The CRA uses an LLM-based debate prompt:

$$\text{LLM}_{\text{CRA}}\big(t, t'\big) \; \rightarrow \; \{\texttt{Agree}, \texttt{Contradict}\}, \tag{11}$$

If $\text{LLM}_{\text{CRA}}$ yields `Contradict`, $t$ is then discarded or queued for manual expert review, depending on the system's confidence.

### 3.10 Evaluator Agents (EA)

Finally, the Evaluator Agents aggregate multiple verification signals and compute global confidence $C(t)$, clarity $Cl(t)$, and relevance $R(t)$ for each triplet $t$.

$$\text{Confidence:} \quad C(t) = \sigma\Big(\sum \alpha_i v_i(t)\Big), \tag{12}$$

$$\text{Clarity:} \quad Cl(t) = \sigma\Big(\sum \beta_j c_j(t)\Big), \tag{13}$$

$$\text{Relevance:} \quad R(t) = \sigma\Big(\sum \gamma_k r_k(t)\Big), \tag{14}$$

where $\sigma(x) = \frac{1}{1+e^{-x}}$ and $\{\alpha_i, \beta_j, \gamma_k\}$ reflect the trustworthiness of each verification source, and $v_i, c_j, r_k$ are verification signals for confidence, clarity, and relevance respectively. We finalize $t$ for integration using the mean score:

$$\text{integrate}(t) = \begin{cases} 1, & \text{if } \frac{C(t)+Cl(t)+R(t)}{3} \geq \Theta \\ 0, & \text{otherwise.} \end{cases} \tag{15}$$

Altogether, this multi-agent pipeline, fully powered by specialized LLMs in each stage, enables robust, scalable, and accurate enrichment of large-scale KG. Future extensions can easily incorporate new domain ontologies, additional specialized agents, or updated LLM prompts as tasks continues to evolve.

## 4 Experimental Setup

This section presents a comprehensive proof-of-concept evaluation settings of the proposed KARMA framework. Unlike conventional NLP tasks that rely on a gold-standard dataset of biomedical entities and relationships, our evaluation adopts a multi-faceted approach. We integrate LLM-based verification with specialized graph-level metrics to assess the quality of the generated knowledge graph. The evaluation spans genomics, proteomics, and metabolomics, showcasing KARMA's adaptability across diverse biomedical domains.

### 4.1 Data Collection

We curate scientific publications from PubMed [27] across three primary domains: the *Genomics Corpus*, which includes 720 papers focused on gene variants, regulatory elements, and sequencing studies; the *Proteomics Corpus*, comprising 360 papers related to protein structures, functions, and protein-interaction networks; and the *Metabolomics Corpus*, containing 120 papers discussing metabolic pathways, metabolite profiling, and clinical applications. All articles are stored in PDF format and processed by the *Ingestion Agent* within KARMA.

### 4.2 LLM Backbones

We evaluate three general-purpose LLMs as the backbone for KARMA's multi-agent knowledge graph enrichment pipeline using their APIs.

*GLM-4* [7]: An open-source 9B-parameter model, achieving 72.4 on the MMLU NLP benchmark.

*GPT-4o* [1]: A proprietary multimodal model optimized through RLHF. It has demonstrated strong adaptability in scientific knowledge extraction and concept grounding [4].

*DeepSeek-v3* [15]: An open-source 37-billion-activated-parameter mixture-of-experts (MoE) model with strong focus on STEM domains.

Each KARMA agent (e.g., *Reader, Summarizer, Extractor*) shares the same LLM backbone per experiment. All LLM-based evaluations employ DeepSeek-v3. Prompting strategies, detailed in Appendix B, are minimally modified to ensure comparability across LLMs and domains. We analyze variations in the final constructed knowledge graph based on different LLM backbones.

### 4.3 Metrics

To evaluate the enriched knowledge graph (KG) in the absence of a gold-standard reference, we employ a multi-faceted evaluation framework that assesses structural integrity, correctness, and practical utility. This framework comprises three categories of metrics: core metrics, graph statistics, and quality indicators. Together, these metrics provide comprehensive insights into the quality and usability of newly added triples and the overall augmented KG.

**Core Metrics** focus on the properties of newly added triples using structural and LLM-based indicators. The *Average Confidence* ($M_{Con}^{\uparrow}$) measures the mean confidence scores across all new triples, reflecting their reliability. The *Average Clarity* ($M_{Cla}^{\uparrow}$) computes the mean clarity scores, indicating how unambiguous or direct each relation is. The *Average Relevance* ($M_{Rel}^{\uparrow}$) captures the

Table 1: KARMA evaluation metrics across domains and models. $M_{Con}^{\uparrow}$: Average confidence score, $M_{Cla}^{\uparrow}$: Average clarity score, $M_{Rel}^{\uparrow}$: Average relevance score, $\Delta_{Cov}^{\uparrow}$: Coverage gain, $\Delta_{Con}^{\uparrow}$: Connectivity gain, $R_{CR}^{\downarrow}$: Conflict ratio, $R_{LC}^{\uparrow}$: LLM-based correctness score, $C_{QA}^{\uparrow}$: QA coherence score, $R_{HE}^{\uparrow}$: Human evaluation score. Best performance in each domain highlighted.

| Domain | Model | Core Metrics | | | Graph Stats. | | Quality Indicators | | | |
|---|---|---|---|---|---|---|---|---|---|---|
| | | $M_{Con}^{\uparrow}$ | $M_{Cla}^{\uparrow}$ | $M_{Rel}^{\uparrow}$ | $\Delta_{Cov}^{\uparrow}$ | $\Delta_{Con}^{\downarrow}$ | $R_{CR}^{\uparrow}$ | $R_{LC}^{\uparrow}$ | $C_{QA}^{\uparrow}$ | $R_{HE}^{\uparrow}$ |
| **Genomics** | Single-Agent | NA | NA | NA | 4384 | 1.083 | NA | 0.493 | 0.472 | 0.320 |
| | GLM-4 | 0.729 | 0.804 | **0.716** | 4969 | 1.131 | 0.238 | 0.623 | 0.589 | 0.445 |
| | GPT-4o | 0.843 | 0.744 | 0.640 | 9795 | 1.265 | **0.148** | **0.880** | 0.569 | 0.510 |
| | DeepSeek-v3 | **0.846** | **0.754** | 0.667 | **38230** | **1.765** | 0.186 | 0.831 | **0.612** | **0.625** |
| **Proteomics** | Single-Agent | NA | NA | NA | 5002 | 1.150 | NA | 0.638 | 0.572 | 0.415 |
| | GLM-4 | 0.731 | 0.752 | 0.609 | 6832 | 1.173 | 0.214 | 0.720 | **0.617** | 0.500 |
| | GPT-4o | 0.823 | 0.797 | 0.613 | 7008 | 1.191 | 0.160 | 0.740 | 0.612 | 0.550 |
| | DeepSeek-v3 | **0.845** | **0.825** | **0.682** | **11936** | **1.468** | **0.151** | **0.772** | 0.613 | **0.575** |
| **Metabolomics** | Single-Agent | NA | NA | NA | 485 | 1.077 | NA | 0.527 | 0.450 | 0.455 |
| | GLM-4 | 0.701 | **0.790** | 0.762 | 703 | 1.159 | 0.188 | 0.617 | 0.449 | 0.485 |
| | GPT-4o | **0.802** | 0.730 | 0.726 | 773 | 1.143 | 0.147 | **0.683** | 0.482 | 0.535 |
| | DeepSeek-v3 | 0.790 | 0.746 | **0.767** | **1752** | **1.811** | **0.132** | 0.668 | **0.493** | **0.580** |

mean relevance scores, assessing the domain significance of the triples. These metrics collectively evaluate the intrinsic quality of the added knowledge.

**Graph Statistics** quantify the structural properties of the augmented KG. The *Coverage Gain* ($\Delta_{Cov}^{\uparrow}$) measures the number of newly introduced entities not previously in the KG, reflecting its expanded scope. The *Connectivity Gain* ($\Delta_{Con}^{\uparrow}$) calculates the net increase in node degrees (summed over existing entities), indicating enhanced interconnectedness.

**Quality Indicators** assess reliability and usability through multiple lenses. The *Conflict Ratio* ($R_{CR}^{\downarrow}$) represents the fraction of newly extracted edges removed by the *ConflictResolutionAgent* due to internal or external contradictions. The *LLM-based Correctness* ($R_{LC}^{\uparrow}$) is determined by a hold-out LLM judging each new triple ((head, r, tail)) as likely correct, uncertain, or likely incorrect, with $R_{LC} = \frac{\#(\text{likely correct})}{\#(\text{all new triples})}$. The Question-Answer Coherence ($C_{QA}^{\uparrow}$) evaluates the fraction of plausible KG-derived answers for a curated set of domain-specific questions answerable via KG traversal. Finally, the *Human Evaluation Score* ($R_{HE}^{\uparrow}$) scaled from 0 to 1, gauges the quality of triple extractions based on assessments by two human experts, offering a comprehensive measure of the knowledge graph's accuracy and utility.

These complementary metrics provide insights into the structural integrity, internal consistency, correctness, and practical utility of the enriched knowledge graph.

## 5 Results

### 5.1 Overall Evaluation

Our comprehensive evaluation (Table 1, with examples in Appendix C.1,C.2,C.3) demonstrates that KARMA significantly extends domain-specific knowledge graphs through its multi-agent architecture. Four key findings emerge: (1) The framework demonstrates superior performance compared to the GLM-4-based single-agent approach, which extracts all triples in a single generation, (2) The framework exhibits varying performance across distinct domains; it identifies the most entities in prevalent fields such as genomics (53.1/article), achieving 3.6× higher coverage gain ($\Delta_{Cov}$) per article than metabolomics (14.6/article); (3) LLM backbone selection substantially impacts KG quality, with DeepSeek-v3 achieving superior performance on 17/24 (71%) metrics across domains;

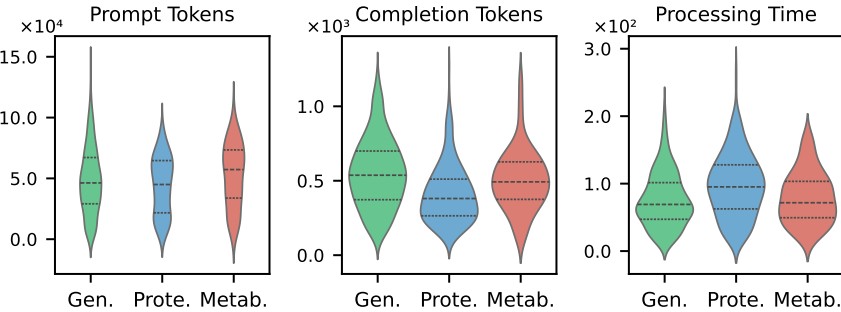

Figure 3: Comparison of prompt tokens, completion tokens, and processing time across domains.

(4) Evaluating knowledge and resolving conflicts automatically can enhance the quality of the extracted knowledge graph, improving LLM-based accuracy by $4.6\%$–$14.4\%$.

## 5.2 Domain-Level Observations.

**Genomics: Scale Meets Precision (C.1)** The genomics domain (720 papers) exhibits the most pronounced model differentiation. DeepSeek-v3 achieves $\Delta_{Cov} = 38,230$ while maintaining a competitive correctness score $R_{LC} = 0.831$, only $5.6\%$ below GPT-4o's peak. This suggests that MoE architectures can balance recall and precision in large-scale extraction.

**Proteomics: Balanced Optimization (C.2)** With 360 papers, proteomics reveals balanced gains: DeepSeek-v3 leads in both core metrics ($M_{Con} = 0.845$) and structural gains ($\Delta_{Con} = 1.468$), while GLM-4 achieves peak QA coherence ($C_{QA} = 0.617$). The $19.1\%$ higher $\Delta_{Cov}$ for DeepSeek-v3 versus GPT-4o indicates greater sensitivity to protein interaction nuances.

**Metabolomics: Specialization Pays Off (C.3)** Despite the smallest corpus (120 papers), GLM-4 delivers superior clarity ($M_{cla} = 0.790$) and GPT-4o excels in correctness ($R_{LC} = 0.683$). However, DeepSeek-v3's $\Delta_{Con} = 1,752$ is $127\%$ higher than GPT-4o, demonstrates unique capability to extrapolate metabolic pathways from limited data.

## 5.3 Analysis of LLM Backbones

Our comparison reveals strengths of different backbones: DeepSeek-v3 drives unparalleled coverage gains, outpacing GPT-4o by $3.9\times$ in genomics and $2.3\times$ in metabolomics while maintaining competitive correctness ($R_{LC} = 0.831$ vs GPT-4o's 0.880 in genomics). This contrasts with GPT-4o's precision-first profile, where it achieves peak $R_{LC}$ scores (0.880 genomics, 0.740 proteomics) but yields $41\%$ lower connectivity gains than DeepSeek-v3, reflecting underutilized implicit relationships. GLM-4, though smaller (10B parameters), demonstrates domain-specific prowess: its biomedical tuning delivers best-in-class metabolomics clarity ($M_{Cla} = 0.762$) and proteomics QA coherence ($C_{QA} = 0.617$), while its conflict ratio ($R_{CR} = 0.188$) remains competitive despite lower parameter count. The tradeoffs (DeepSeek-v3's coverage balance for correctness, GPT-4o's precision sacrifice for completeness, GLM-4's niche adaptation) underscore why *KARMA*'s multi-agent framework strategically decouples extraction, validation, and can utilize the strengths of each backbone. Different backbones also lead to variations in the distribution of key evaluation metrics (Figure C.1,C.2,C.3).

## 5.4 Cost Analysis

The evaluation of computational costs (Figure 3) demonstrates distinct trade-offs in token usage and processing time across different domains. The variations in article lengths and information density naturally lead to differences in token consumption and processing times. Notably, genomics shows higher completion token distributions (mean = 550.64, std = 232.92), explaining KARMA's higher $\Delta_{Cov}$ in this domain. Meanwhile, proteomics exhibits broader processing time distributions (mean = 96.58, std = 46.90), which correlates with its stronger performance in knowledge quality metrics ($R_{LC}$ and $C_{QA}$), suggesting that longer processing times contribute to more thorough relationship analysis and validation.

Table 2: Ablation study results for KARMA, evaluating the impact of different agents (Summarizer, Conflict Resolution, Evaluator).

| Domain | Configuration | Quality Indicators | | | |
|--------|---------------|------------|------------|------------|------------|
| | | $R_{\text{CR}}^{\uparrow}$ | $R_{\text{LC}}^{\uparrow}$ | $C_{\text{QA}}^{\uparrow}$ | $R_{\text{HE}}^{\uparrow}$ |
| **Genomics** | w/o Summarizer Agent | 0.758 | 0.788 | 0.472 | 0.600 |
| | w/o Conflict Resolution Agent | 0.790 | 0.733 | 0.554 | 0.485 |
| | w/o Evaluator Agent | 0.793 | 0.752 | 0.561 | 0.515 |
| | KARMA-Full | **0.831** | **0.831** | **0.612** | **0.625** |
| **Proteomics** | w/o Summarizer Agent | 0.632 | 0.759 | 0.547 | 0.610 |
| | w/o Conflict Resolution Agent | 0.661 | 0.742 | 0.583 | 0.525 |
| | w/o Evaluator Agent | 0.696 | 0.755 | 0.605 | 0.580 |
| | KARMA-Full | **0.772** | **0.772** | **0.613** | **0.625** |
| **Metabolomics** | w/o Summarizer Agent | 0.577 | 0.592 | **0.537** | 0.555 |
| | w/o Conflict Resolution Agent | 0.629 | 0.551 | 0.471 | 0.545 |
| | w/o Evaluator Agent | 0.603 | 0.572 | 0.480 | 0.550 |
| | KARMA-Full | **0.668** | **0.608** | 0.493 | **0.580** |

## 5.5 Ablation Study

To better quantify the contributions of each specialized agent in KARMA, we conduct an ablation study (Table 2) by systematically removing or replacing selected agents and measure the resulting performance across the three domains. Specifically, we evaluate:

- **KARMA-Full:** All agents active, including Summarizer, Conflict Resolution, and Evaluator modules.
- **w/o Summarizer**: Bypasses the Summarizer Agents, passing all text directly from Reader Agents to Entity and Relationship Extraction.
- **w/o Conflict Resolution**: Disables the Conflict Resolution Agent, allowing potentially contradictory edges into the final graph.
- **w/o Evaluator**: Omits the final confidence, clarity, and relevance evaluation and aggregation, integrating relationships without filtering.

We conduct these ablations using the same LLM backbone (DeepSeek-v3 in our experiments) for consistency. Table 2 summarizes the impact on evaluation metrics ($R_{LC}$, $C_{QA}$) for each domain. The ablation study highlights the importance of each agent in KARMA's performance. Removing the *Summarizer Agent* produce much more entities and triples, but reduces accuracy ($C_{QA}$ drop 22.9% ($0.612 \rightarrow 0.472$) in genomics) and coherence ($R_{LC}$ drop 18.2% ($0.772 \rightarrow 0.632$) in proteomics), as unfiltered text introduces noise. Disabling the *Conflict Resolution Agent* significantly lowers correctness ($C_{QA}$ drop 4.9% ($0.831 \rightarrow 0.790$) in genomics), especially in resolving contradictions like conflicting gene-disease associations. Omitting the *Evaluator Agents* has the most impact on usability, as unfiltered, low-confidence edges degrade answer quality ($R_{LC}$ drop 9.7% ($0.668 \rightarrow 0.603$) in metabolomics). Across all domains, conflict resolution proves critical for maintaining logical consistency, while summarization and evaluation ensure focused extraction and high-quality integration. This demonstrates that KARMA's multi-agent design is essential for balancing accuracy, consistency, and usability in KG enrichment.

## 6 Conclusion

We introduce KARMA, a multi-agent LLM framework designed to tackle the challenge of scalable knowledge graph enrichment from scientific literature. By decomposing the extraction process into specialized agents for entity discovery, relationship validation, and conflict resolution, KARMA ensures adaptive and accurate knowledge integration. Its modular design reduces the impact of conflicting edges through multi-layered assessments and cross-agent verification. Experimental results across genomics, proteomics, and metabolomics demonstrate that multi-agent collaboration can overcome the limitations of single-agent approaches, particularly in domains that require complex semantic understanding and adherence to structured schemas.

## Acknowledge

This research was supported by National Key Research and Development Program of China (2024YFF0507400) and National Natural Science Foundation of China (6220071694).

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

# Appendix

# A  Limitations

Despite the promising performance of KARMA, several limitations remain. First, our evaluation relies primarily on LLM-based metrics rather than direct human expert validation. While we employ multi-faceted metrics (e.g., QA coherence, conflict resolution) to assess the quality of the extracted knowledge, we recognize that domain experts must ultimately verify critical biomedical claims before applying them in clinical settings. Furthermore, performance varies across domains; for instance, metabolomics shows 12.4% and 11.9% lower QA coherence than proteomics and genomics, respectively, indicating challenges in modeling sparse and rare relationships in this field. These limitations highlight opportunities for future improvements, such as integrating hybrid neuro-symbolic approaches and optimizing agent coordination protocols.

# B  Detailed propmts for KARMA agents

This appendix provides example prompts for each agent in the KARMA framework. All agents operate via LLMs with specialized prompt templates. We emphasize confidence, clarity, and domain relevance. Where applicable, we include sample inputs, outputs, and negative examples to illustrate how each agent handles complexities in the context.

## B.1  Function summaries of different agents

The KARMA framework comprises nine specialized LLM-powered agents, each handling distinct stages of the knowledge extraction and integration task. Below are their core functions:

- **Ingestion Agents (IA)**: Retrieve raw documents (PDF/HTML), normalize text (handling OCR errors, tables), and extract metadata (authors, journal, publication date).
- **Reader Agents (RA)**: Split documents into sections, score segment relevance using KG context, and filter non-revelant content (e.g., acknowledgments).
- **Summarizer Agents (SA)**: Condense text segments into concise summaries while preserving entity relationships (e.g., "Drug X inhibits Protein Y, reducing Disease Z symptoms" → "X inhibits Y; Y linked to Z").
- **Entity Extraction Agents (EEA)**: Identify entities via few-shot LLM prompts, then normalize them to KG canonical forms using ontology-guided embedding alignment.
- **Relationship Extraction Agents (REA)**: Detect relationships (e.g., `treats`, `causes`) between entity pairs using multi-label classification, allowing overlapping relations (e.g., "Drug A both `inhibits` Protein B and `triggers` Side Effect C").
- **Schema Alignment Agents (SAA)**: Map novel entities/relations to KG schema types (e.g., classifying "CRISPR-Cas9" as `Gene-Editing Tool`) or flag them for ontology expansion.
- **Conflict Resolution Agents (CRA)**: Resolve contradictions (e.g., new triplet "Drug D `treats` Disease E" vs. existing "Drug D `exacerbates` Disease E") via LLM debate and evidence aggregation.
- **Evaluator Agents (EA)**: Compute integration confidence using weighted signals (confidence, relevance, clarity) and apply threshold-based final approval.

## B.2  Additional Notes on Prompt Engineering

### 1. Example-based prompting (few-shot).

In practice, each agent's prompt can be extended with short examples of input-output pairs to provide the LLM with more context, thereby improving the accuracy and consistency of its responses. For instance, the EEA prompt might include examples of drug-disease pairs, while the CRA prompt might illustrate how to handle partial contradictions vs. direct contradictions.

### 2. Negative Examples and Error Correction.

To increase robustness, each agent can be provided with negative examples or clarifications on error-prone cases. For example, the Summarizer Agent might be shown how not to remove important numerical dosage information; the EEA might have a demonstration of ignoring location references that are not topic-related entities (e.g., "Paris" is not a `Disease`).

**3. Incremental Fine-Tuning and Updates.**

As knowledge evolves, so do the vocabularies and relationship types. Agents can be periodically re-trained or their prompts updated to handle newly emerging entities (e.g., novel viruses, new drug classes) and complex multi-modal relationships. The modular structure of the prompts eases integration of these updates without redesigning the entire pipeline.

Collectively, these prompts enable KARMA to harness LLMs at every stage of the knowledge extraction and integration process, resulting in a dynamic, scalable, and accurate knowledge enrichment.

## B.3 Ingestion Agent (IA) Prompt

**Title:** *IA_Prompt*

**Role Description:** You are a clinical expert. Your responsibility is to: 1. Retrieve raw publications from designated sources (e.g., PubMed, internal repositories). 2. Convert various file formats (PDF, HTML, XML) into a consistent normalized text format. 3. Extract metadata such as the title, authors, journal/conference name, publication date, and unique identifiers (DOI, PubMed ID).

**System Instruction:**

- **Input:** Raw document payload or a path/URL to the document, plus minimal metadata (if available).

- **Output:** A JSON structure with two main fields:

    1. `"metadata"`: {title, authors, journal, pub_date, doi, pmid, etc.}
    2. `"content"`: A single string or a structured array containing the full text, preserving headings or major sections if possible.

- **Key Requirements:**

    – Handle OCR artifacts if the PDF is scanned (e.g., correct typical OCR errors where possible).
    – Normalize non-ASCII characters (greek letters, special symbols) to ASCII or minimal LaTeX markup when relevant (e.g., \alpha).
    – If certain fields cannot be extracted, leave them as empty or `"N/A"` but do not remove the key from the JSON.

- **Error Handling:**

    – In case of partial or unreadable text, mark the corrupted portions with place-holders (e.g., "[UNREADABLE]").
    – If the document is locked or inaccessible, set an error flag in the output JSON.

**LLM Prompt Template (Illustrative Example):**
```
[System Role:  IngestionAgent]
You will receive a raw publication in PDF or HTML format.
1.  Extract all available metadata:  Title, Authors, Date,
Journal/Source, PMID, DOI.
2.  Convert the text to ASCII or minimal LaTeX.
3.  Provide a JSON output with keys:  {"metadata":  {...}, "content":
"..." }.
4.  If any portion of the text is unreadable, replace it with
"[UNREADABLE]".
```

**Sample Input:**
```
pdf_document:  "Binary PDF data...", doi:  "10.1000/j.jmb.2022.07.123"
```

**Sample Output:**
```
{
"metadata":  {"title":  "Novel Anti-viral Therapy", "authors":  ["Jane
Doe"],
"content":  "Introduction
n Recent advances in...  Methods
n We tested..."
}
```

## B.4 Reader Agent (RA) Prompt

**Title:** *RA_Prompt*

**Role Description:** You are the **Reader Agent**. Your goal is to parse the normalized text from the IA and generate *logical segments* (e.g., paragraph-level chunks) that are likely to contain relevant knowledge. Each segment must be accompanied by a numeric *Relevance Score* indicating its importance for downstream extraction tasks.

**System Instruction:**

- **Input:** JSON output from IA with `"metadata"` and `"content"` fields.
- **Output:** A JSON array `"segments"`, where each element is {"text": "...", "score": 0.xxx}.
- **Scoring Heuristics:**
  - Use domain knowledge (e.g., presence of known keywords, synonyms, or known entity patterns) to increase the score.
  - Use structural cues (e.g., headings like "Results", "Discussion" might have higher relevance for new discoveries).
  - If a segment is purely methodological (e.g., protocols or references to equipment) with no new knowledge, assign a lower score.
- **Edge Cases:**
  - Very short segments (<30 characters) or references sections might be assigned a minimal score.
  - If certain sections are incomplete or corrupted, still generate a segment but label it with `"score": 0.0`.

**LLM Prompt Template (Illustrative Example):**

```
[System Role: ReaderAgent]
Given the JSON with metadata and a large text string under "content",
split the text into smaller segments (e.g., paragraphs).
For each segment, estimate a Relevance Score (0 to 1) that indicates
the likelihood of containing novel relationships or key findings.
Output a JSON array "segments": [{"text":"...", "score": 0.XX},
...].
```

**Sample Input:**

```
{ "metadata": {"title": "Antimicrobial Study"...},
"content": "Abstract
n We tested new...
n Methods
n The protocol was...
n"
}
```

**Sample Output:**

```
{ "segments": [
{"text": "Abstract We tested new...", "score": 0.85},
{"text": "Methods The protocol was...", "score": 0.30}]
}
```

## B.5 Summarizer Agent (SA) Prompt

**Title:** *SA_Prompt*

**Role Description:** You are the **Summarizer Agent**. Your task is to convert high-relevance segments into concise summaries while retaining technical detail such as gene symbols, chemical names, or numeric data that may be crucial for entity/relationship extraction.

**System Instruction:**

- **Input:** A set of segments, each with a relevance score (e.g., from the RA).
- **Output:** A JSON array `"summaries"`, each entry with:
    1. `"original_text"`: the original segment
    2. `"summary"`: a concise, domain-specific summary (2–4 sentences recommended)
    3. `"score"`: the inherited or slightly adjusted relevance score
- **Summarization Rules:**
    - Avoid discarding domain-specific terms that could indicate potential relationships. For example, retain "IL-6" or "p53" references precisely.
    - If numeric data is relevant (e.g., concentrations, p-values), incorporate them verbatim if possible.
    - Keep the summary length under 100 words to reduce computational overhead for downstream agents.
- **Handling Irrelevant Segments:**
    - If the Relevance Score is below a threshold (e.g., 0.2), you may skip or heavily compress the summary.
    - Mark extremely low relevance segments with `"summary":   "[OMITTED]"` if not summarizable.

**LLM Prompt Template (Illustrative Example):**
```
[System Role:  SummarizerAgent]
For each segment with (text, score), produce a summary capturing key
biomedical elements (drugs, diseases, molecular targets).
Preserve numeric data or specific chemical/gene names.  Output a JSON
list:
{"summaries":  [{"original_text":  "...", "summary":  "...", "score":
0.xx}, ...]}.
```
**Sample Input:**
```
{ "segments":   [
{"text":   "In this study, IL-6 blockade ...", "score":  0.90},
{"text":   "The control group had p=0.01...", "score":  0.75}]
}
```
**Sample Output:**
```
{ "summaries":   [
{"original_text":"In this study, IL-6 blockade...",
 "summary":"This segment describes the use of IL-6 inhibition in ...",
 "score":  0.90},
{"original_text":"The control group had p=0.01...",
 "summary":"Researchers observed a statistically significant difference
(p=0.01) between ...", "score":  0.75}]
}
```

## B.6 Entity Extraction Agent (EEA) Prompt

**Title:** *EEA_Prompt*

**Role Description:** You are the **Entity Extraction Agent**. Based on summarized text, your objective is to: 1. Identify biomedical entities (Disease, Drug, Gene, Protein, Chemical, etc.). 2. Link each mention to a canonical ontology reference (e.g., UMLS, MeSH, SNOMED CT).

**System Instruction:**

- **Input:** A summarized text from the SA outputs.
- **Output:** JSON `"entities"` array, where each element includes:
    1. `"mention"`: the exact substring from the text
    2. `"type"`: e.g., `"Drug"`, `"Disease"`, `"Gene"`, etc.
    3. `"normalized_id"`: references such as `"UMLS:C0004238"` or `"MESH:D001943"`
- **LLM-driven NER:**
    - Use domain-specific knowledge to identify synonyms ("acetylsalicylic acid" → Aspirin).
    - Include multi-word expressions ("breast cancer" as a single mention).
- **Handling Ambiguity:**
    - If multiple ontology matches are possible, list the top candidate plus a short reason or partial mention of the second-best match.
    - If no suitable ontology reference is found, set `"normalized_id":` `"N/A"` and keep the raw mention.

**LLM Prompt Template (Illustrative Example):**
```
[System Role:  EntityExtractorAgent]
Identify all biomedical entities from the text snippet.  Output array
"entities":  [{"mention":"...", "type":"...", "normalized_id":"..."},
...].
Use domain ontologies (UMLS, MeSH, SNOMED) to map the mention to a
canonical identifier if possible.
```

**Sample Input:**
```
{ "summary":  "We tested Aspirin for headache relief at a dosage of 100
mg."
}
```

**Sample Output:**
```
{ "entities":  [
{"mention":  "Aspirin", "type":  "Drug", "normalized_id":
"MESH:D001241"},
{"mention":  "headache", "type":  "Disease", "normalized_id":
"UMLS:C0018681"}]
}
```

## B.7 Relationship Extraction Agent (REA) Prompt

**Title:** *REA_Prompt*

**Role Description:** You are the **Relationship Extraction Agent**. Given a text snippet plus a set of recognized entities, your mission is to detect possible relationships (e.g., `treats`, `causes`, `interactsWith`, `inhibits`).

**System Instruction:**

- **Input:** A summary $u_j$ and a list of entity with normalized IDs from the EEA.
- **Output:** A JSON array `"relationships"` where each element is:
    1. `"head"`: the head entity
    2. `"relation"`: the relationship type (string)
    3. `"tail"`: the tail entity
- **LLM-based Relation Classification:**
    - Consider grammar structures (e.g., "X was observed to inhibit Y") and domain patterns ("X reduces expression of Y").
    - Allow multiple relationship candidates if the text is ambiguous or suggests multiple interactions.
- **Negative Relation Handling:**
    - If the text says "Aspirin *does not* treat migraine," the relationship (`Aspirin`, `treats`, `migraine`) is negative. Output either no relationship or a negative-labeled relationship (implementation-specific).
    - Recognize negation cues ("no effect", "absence of association").

**LLM Prompt Template (Illustrative Example):**
```
[System Role:  RelationshipExtractorAgent]
You will receive text along with extracted entities.  Determine if any
pair of entities has a meaningful relationship.  Use domain knowledge
to find patterns like "X treats Y", "X inhibits Y", etc.
Output each discovered relationship with "head", "relation", "tail",
and "confidence".
```

**Sample Input:**
```
{      "summary":  "Aspirin was shown to reduce headaches by inhibiting
prostaglandin...",
"entities":  [{"mention":  "Aspirin", "normalized_id":
"MESH:D001241"},
{"mention":  "headaches", "normalized_id":  "UMLS:C0018681"},
{"mention":  "prostaglandin", "normalized_id":  "MESH:D011441"}]
}
```

**Sample Output:**
```
{ "relationships":  [
{"head":  "MESH:D001241", "relation":  "treats", "tail":
"UMLS:C0018681"},
{"head":  "MESH:D001241", "relation":  "inhibits", "tail":
"MESH:D011441"}]
}
```

## B.8 Schema Alignment Agent (SAA) Prompt

**Title:** *SAA_Prompt*

**Role Description:** You are the **Schema Alignment Agent**. Newly extracted entities or relationships may not match existing KG classes or relation types. Your job is to determine how they should map onto the existing ontology or schema.

**System Instruction:**

- **Input:** A list of new entities or relations that appear in the extraction but are not recognized in the current KG schema.

- **Output:** An array `"alignments"` with objects {`"id"`:..., `"type"`:..., `"status"`:...}, possibly plus a `"new_types"` array for unrecognized patterns.

- **Ontology Reference:**
    - For each unknown entity, propose a parent type from {`Drug`, `Disease`, `Gene`, `Chemical`, ...} if not in the KG.
    - For each unknown relation, map it to an existing relation if semantically close. Otherwise, propose a new label.

- **Confidence Computation:**
    - Consider lexical similarity, embedding distance, or domain rules (e.g., if an entity ends with "-in" or "-ase", it might be a protein or enzyme).
    - Provide a final numeric score for how certain you are of the proposed alignment.

**LLM Prompt Template (Illustrative Example):**
```
[System Role:  SchemaAlignmentAgent]
You will receive a list of new entities/relations that are not in the
KG. Try mapping them to existing node/edge types.
Output JSON: {"alignments":[{"id":"...", "proposed_type":"...",
"status":"mapped"/"new"},...]}.
```
**Sample Input:**
```
{ "unknown_entities":  ["TNF-alpha", "miR-21"],
"unknown_relations":  ["overexpresses"]
}
```
**Sample Output:**
```
{ "alignments":  [
{"id":"TNF-alpha", "proposed_type":"Protein", "status":"mapped"},
{"id":"miR-21", "proposed_type":"RNA", "status":"new"}],
 "new_relations":[
{"relation":"overexpresses", "closest_match":"upregulates","status":"new"}]
}
```

## B.9  Conflict Resolution Agent (CRA) Prompt

**Title:** *CRA_Prompt*

**Role Description:** You are the **Conflict Resolution Agent**. Sometimes new triplets are detected that contradict existing knowledge (e.g., (DrugX, causes, DiseaseY) vs. (DrugX, treats, DiseaseY)). Your role is to classify these into `Contradict`, `Agree`, or `Ambiguous`, and decide whether the new triplet should be discarded, flagged for expert review, or integrated with caution.

**System Instruction:**

- **Input:** A new candidate triplet $t$ and a potentially conflicting triplet $t'$ already in the KG.
- **Output:** A JSON object with:
    1. "decision": "Contradict", "Agree", or "Ambiguous"
    2. "resolution": {"action": "discard"/"review"/"integrate", "rationale": "..."}
- **LLM-based Debate**:
    - Use domain knowledge to see if relationships can coexist (e.g., `inhibits` vs. `activates` are typically contradictory for the same target).
    - Consider partial contexts, e.g., different dosages or subpopulations.
- **Escalation Criteria**:
    - If the new triplet has high confidence but conflicts with old data that has lower confidence, consider overriding or `review`.
    - If both are high confidence, label `Contradict`, prompt manual verification.

**LLM Prompt Template (Illustrative Example):**
```
[System Role:  ConflictResolutionAgent]
You have two triplets t_new and t_existing that appear to conflict.
Determine if they truly contradict or if they could be contextually
compatible.
Output {"decision":"Contradict" / "Agree" / "Ambiguous",
"resolution":{"action":"discard" / "review" / "integrate",
"rationale":"..."}}.
```

**Sample Input:**
```
{ "t_new":  {"head":"DrugX",
"relation":"treats", "tail":"DiseaseY"},
"t_existing":  {"head":"DrugX", "relation":"causes",
"tail":"DiseaseY"} }
```

**Sample Output:**
```
{ "decision":"Contradict",
"resolution":  {"action":"review", "rationale":"Both have high
confidence; manual verification required."} }
```

## B.10   Evaluator Agent (EA) Prompt for Confidence

**Title:** *EA_Prompt_confidence*

**Role Description:** You are the **Evaluator Agent**. After the extraction, alignment, and conflict resolution phases, each candidate triplet has multiple verification scores from external databases, additional LLM-based checks, or domain-specific classifiers. Your duty is to aggregate these signals into a final confidence score $C(t)$ and decide whether to integrate each triplet into the KG.

**System Instruction:**

- **Input:** A list of triplets, each with:
    1. Partial confidence scores (e.g., $v_1, v_2, ..., v_N$).
    2. Conflict resolution status (`"Contradict"`, `"Agree"`, or `"Ambiguous"`).
- **Output:** A JSON array `"final_triplets"` with:
    1. `"head"`, `"relation"`, `"tail"`: identifiers for the triplet
    2. `"final_confidence"`: combined confidence score $C(t)$
- **Aggregation Formula:**
    - You must also factor in conflict resolution outcomes: if `Contradict`, $C(t)$ is penalized or forced to 0 unless manual override occurs.

**LLM Prompt Template (Illustrative Example):**
```
[System Role:  EvaluatorAgent]
Given an array of triplets with partial scores [v1, v2, ...], conflict
status, etc., compute a final confidence using logistic weighting.
Output as {"final_triplets":  [{"head":..., "relation":...,
"tail":..., "final_confidence":...}, ...]}.
```
**Sample Input:**
```
{ "candidates":  [
{"head":"MESH:D001241","relation":"treats", "tail":"UMLS:C0018681",
"scores":  [0.90, 0.85], "conflict":"Agree"},
{"head":"DrugX","relation":"causes", "tail":"DiseaseY",
"scores":  [0.70, 0.60], "conflict":"Contradict"}]
}
```
**Sample Output:**
```
{ "final_triplets":  [
{"head":"MESH:D001241","relation":"treats", "tail":"UMLS:C0018681",
"final_confidence":0.87},
{"head":"DrugX","relation":"causes", "tail":"DiseaseY",
"final_confidence":0.65}]
}
```

## B.11 Evaluator Agent (EA) Prompt for Clarity

**Title:** *EA_Prompt_clarity*

**Role Description:** You are the **Evaluator Agent** responsible for assessing the **clarity** of each candidate triplet. After the initial extraction, some triplets may contain ambiguous terminology or uncertain references. Your job is to assign a clarity score $Cl(t)$ to each triplet and decide whether it is sufficiently clear to be integrated into the Knowledge Graph (KG).

**System Instruction:**

- **Input:** A list of triplets, each with:
    1. Partial clarity metrics (e.g., $c_1, c_2, ..., c_N$) obtained from lexical or semantic checks.
    2. A note on whether the triplet's terms or relation are ambiguous, e.g., `"AmbiguousTerm"`, `"ClearTerm"`, etc.
- **Output:** A JSON array `"final_triplets"` with:
    1. `"head"`, `"relation"`, `"tail"`: identifiers for the triplet.
    2. `"final_clarity"`: the combined clarity score $Cl(t)$.

- **Aggregation Formula:**
    - You may apply a weighted averaging or logistic function over $c_1, c_2, ..., c_N$.
    - Downweight or penalize triplets tagged as having ambiguous or unclear terms.

**LLM Prompt Template (Illustrative Example):**
```
[System Role:  EvaluatorAgent]
Given an array of triplets with partial clarity metrics [c1, c2, ...]
and any notes on ambiguity, compute a final clarity score.
Output as {"final_triplets":  [{"head":..., "relation":...,
"tail":..., "final_clarity":...}, ...]}.
```
**Sample Input:**
```
{ "candidates":  [
{"head":"DrugA","relation":"may_treat", "tail":"ConditionB",
"clarity_metrics":  [0.80, 0.85], "ambiguous":"False"},
{"head":"EntityX","relation":"unknownRel", "tail":"EntityY",
"clarity_metrics":  [0.40], "ambiguous":"True"}]
}
```
**Sample Output:**
```
{ "final_triplets":  [
{"head":"DrugA","relation":"may_treat", "tail":"ConditionB",
"final_clarity":0.82},
{"head":"EntityX","relation":"unknownRel", "tail":"EntityY",
"final_clarity":0.40}]
}
```

## B.12 Evaluator Agent (EA) Prompt for Relevance

**Title:** *EA_Prompt_relevance*

**Role Description:** You are the **Evaluator Agent** focusing on the **relevance** of each triplet to the target Knowledge Graph (KG). Some triplets may be factually correct but not pertinent to the KG's domain or scope. Your duty is to compute a relevance score $R(t)$ for each triplet and decide if it should be included in the KG.

**System Instruction:**

- **Input:** A list of triplets, each with:
    1. Partial relevance scores (e.g., $r_1, r_2, ..., r_N$) based on domain-specific criteria (e.g., "medical relevance" or "chemical relevance").
    2. Metadata or tags indicating alignment with the KG's domain (e.g., `"domainMatch"` or `"domainMismatch"`).
- **Output:** A JSON array `"final_triplets"` with:
    1. `"head"`, `"relation"`, `"tail"`.
    2. `"final_relevance"`: the combined relevance score $R(t)$.
- **Aggregation Formula:**
    - Combine the partial relevance scores via an average or logistic function.
    - Penalize triplets flagged as outside of the domain or referencing unknown entities.

**LLM Prompt Template (Illustrative Example):**

```
[System Role:  EvaluatorAgent]
Given an array of triplets with partial relevance scores [r1, r2, ...]
and domain tags, compute a final relevance score.
Output as {"final_triplets":  [{"head":..., "relation":...,
"tail":..., "final_relevance":...}, ...]}.
```

**Sample Input:**

```
{ "candidates":  [
{"head":"DrugA","relation":"used_for", "tail":"DiseaseB",
 "relevance_scores":  [0.90, 0.88], "domainMatch":true},
{"head":"HistoricalFigure","relation":"lived_in",
 "tail":"AncientPlace",
 "relevance_scores":  [0.50], "domainMatch":false}]
}
```

**Sample Output:**

```
{ "final_triplets":  [
{"head":"DrugA","relation":"used_for", "tail":"DiseaseB",
 "final_relevance":0.89},
{"head":"HistoricalFigure","relation":"lived_in",
 "tail":"AncientPlace",
 "final_relevance":0.50}]
}
```

# C   Examples of extracted knowledge graphs

## C.1   Knowledge graph from Genomics articles (Generate using GPT-4o, Examples)

---

### Genomics Knowledge Graph Triples

**Key:** Conf = Confidence, Rel = Relevance, Clr = Clarity

- **EGFR** $\xrightarrow{\text{causes}}$ **lung adenocarcinoma** (Conf: 0.75, Rel: 0.40, Clr: 0.60)
- **EGFR** $\xrightarrow{\text{causes}}$ **non - small cell lung cancer** (Conf: 0.85, Rel: 0.30, Clr: 0.70)
- **T790M** $\xrightarrow{\text{causes}}$ **lung adenocarcinoma** (Conf: 0.98, Rel: 0.20, Clr: 0.70)
- **T790M** $\xrightarrow{\text{causes}}$ **non - small cell lung cancer** (Conf: 0.95, Rel: 0.20, Clr: 0.80)
- **EGFR** $\xrightarrow{\text{activates}}$ **EG** (Conf: 0.75, Rel: 0.20, Clr: 0.99)
- **EGFR** $\xrightarrow{\text{causes}}$ **proliferation** (Conf: 0.85, Rel: 0.50, Clr: 0.60)
- **MTX - 531** $\xrightarrow{\text{treats}}$ **HNSCC** (Conf: 0.99, Rel: 0.80, Clr: 0.50)
- **MTX - 531** $\xrightarrow{\text{used\_in}}$ **PDX** (Conf: 0.75, Rel: 0.80, Clr: 0.50)
- **PIK3CA mutations** $\xrightarrow{\text{associated\_with}}$ **HNSCC** (Conf: 0.85, Rel: 0.70, Clr: 0.99)
- **MTX - 531** $\xrightarrow{\text{treats}}$ **PIK3CA mutations** (Conf: 0.65, Rel: 0.40, Clr: 0.99)
- **MCLA - 158** $\xrightarrow{\text{targets}}$ **EGFR** (Conf: 0.75, Rel: 0.30, Clr: 0.99)
- **EGFR** $\xrightarrow{\text{interacts\_with}}$ **MCLA - 158** (Conf: 0.75, Rel: 0.20, Clr: 0.80)
- **PDOs** $\xrightarrow{\text{used\_in}}$ **biobank** (Conf: 0.75, Rel: 0.50, Clr: 0.50)
- **Retinoic acid** $\xrightarrow{\text{disrupts}}$ **autocrine growth pathway** (Conf: 0.75, Rel: 0.80, Clr: 0.60)
- **Retinoic acid** $\xrightarrow{\text{interacts\_with}}$ **nuclear retinoic acid receptors** (Conf: 0.95, Rel: 0.80, Clr: 0.99)
- **nuclear retinoic acid receptors** $\xrightarrow{\text{regulate}}$ **gene transcription** (Conf: 0.95, Rel: 0.70, Clr: 0.50)
- **EGFR** $\xrightarrow{\text{interacts\_with}}$ **Cys797** (Conf: 0.95, Rel: 0.80, Clr: 0.99)

---

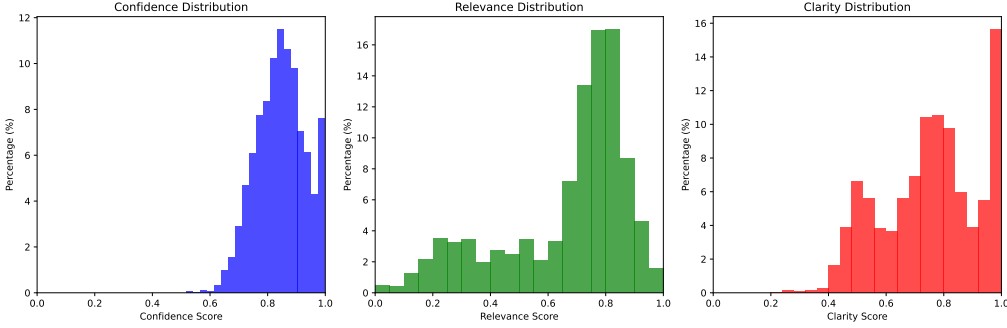

Figure 4: Distribution of confidence, relevance, and clarity scores of extracted genomics knowledge graph triples from KARMA.

## C.2 Knowledge graph from Proteomics articles (Generate using DeepSeek-V3, Examples)

---

### Proteomics Knowledge Graph Triples

**Key:** Conf = Confidence, Rel = Relevance, Clr = Clarity

- **p53** $\xrightarrow{\text{induces}}$ **cell-cycle arrest** (Conf: 0.95, Rel: 0.70, Clr: 0.85)
- **p53** $\xrightarrow{\text{induces}}$ **apoptosis** (Conf: 0.95, Rel: 0.70, Clr: 0.85)
- **mutant p53** $\xrightarrow{\text{causes}}$ **chemotherapy resistance** (Conf: 0.85, Rel: 0.85, Clr: 0.85)
- **MDM2** $\xrightarrow{\text{interacts\_with}}$ **p53** (Conf: 0.95, Rel: 0.20, Clr: 0.90)
- **PRIMA-1** $\xrightarrow{\text{induces}}$ **apoptosis** (Conf: 0.85, Rel: 0.70, Clr: 0.85)
- **NOS2** $\xrightarrow{\text{associated\_with}}$ **cancers** (Conf: 0.85, Rel: 0.70, Clr: 0.60)
- **NOS2 inhibitors** $\xrightarrow{\text{inhibits}}$ **NOS2** (Conf: 0.95, Rel: 0.80, Clr: 0.90)
- **NOS2 inhibitors** $\xrightarrow{\text{reduces}}$ **tumor growth** (Conf: 0.85, Rel: 0.75, Clr: 0.85)
- **NO** $\xrightarrow{\text{induces}}$ **VEGF** (Conf: 0.78, Rel: 0.70, Clr: 0.70)
- **NO** $\xrightarrow{\text{induces}}$ **neovascularization** (Conf: 0.45, Rel: 0.70, Clr: 0.75)
- **NOS2 inhibitors** $\xrightarrow{\text{has\_therapeutic\_potential}}$ **p53-mutant cancers** (Conf: 0.78, Rel: 0.75, Clr: 0.85)
- **tumor progression** $\xrightarrow{\text{dependent\_on}}$ **p53** (Conf: 0.85, Rel: 0.70, Clr: 0.85)
- **NOS2** $\xrightarrow{\text{promotes}}$ **tumor growth** (Conf: 0.85, Rel: 0.85, Clr: 0.75)
- **NOS2** $\xrightarrow{\text{produces}}$ **nitric oxide (NO)** (Conf: 0.95, Rel: 0.90, Clr: 0.90)
- **hypoxia** $\xrightarrow{\text{regulates}}$ **iNOS expression** (Conf: 0.85, Rel: 0.70, Clr: 0.85)
- **iNOS expression** $\xrightarrow{\text{influences}}$ **endothelial integrity** (Conf: 0.85, Rel: 0.70, Clr: 0.75)
- **sulindac sulfide** $\xrightarrow{\text{treats}}$ **cancers** (Conf: 0.78, Rel: 0.70, Clr: 0.75)

---

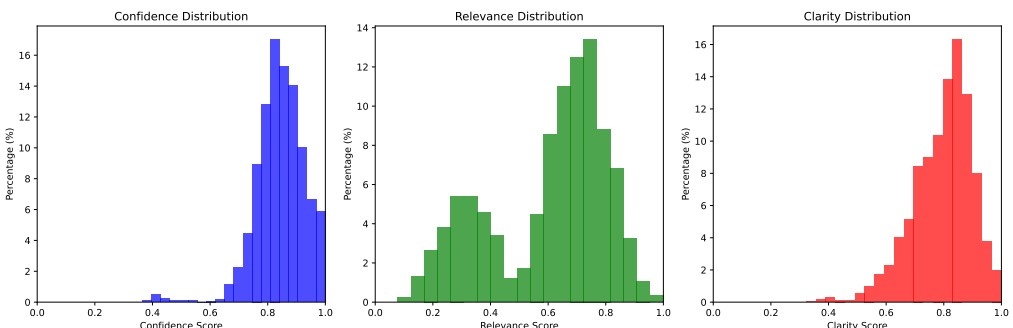

Figure 5: Distribution of confidence, relevance, and clarity scores of extracted proteomics knowledge graph triples from KARMA.

## C.3 Knowledge graph from Metabolomics articles (Generate using GLM-4, Examples)

---

**Metabolomics Knowledge Graph Triples**

**Key:** Conf = Confidence, Rel = Relevance, Clr = Clarity

- **G6PD** $\xrightarrow{\text{activates}}$ **NADPH** (Conf: 0.75, Rel: 0.80, Clr: 0.50)
- **BAG3** $\xrightarrow{\text{interacts\_with}}$ **phosphorylation** (Conf: 0.75, Rel: 0.50, Clr: 0.99)
- **Mitotic NADPH surge** $\xrightarrow{\text{prevents}}$ **chromosome missegregation** (Conf: 0.75, Rel: 0.80, Clr: 0.80)
- **High BAG3 phosphorylation** $\xrightarrow{\text{associated\_with}}$ **poor prognosis** (Conf: 0.75, Rel: 0.80, Clr: 0.99)
- **G6PD** $\xrightarrow{\text{crucial in}}$ **pentose phosphate pathway** (Conf: 0.85, Rel: 0.90, Clr: 0.80)
- **Acetylation at lysine residue K89** $\xrightarrow{\text{activates}}$ **G6PD** (Conf: 0.75, Rel: 0.80, Clr: 0.99)
- **Acetylation at lysine residue K403** $\xrightarrow{\text{inhibits}}$ **G6PD** (Conf: 0.75, Rel: 0.80, Clr: 0.80)
- **astrocyte-to-neuron H2O2 signaling** $\xrightarrow{\text{activates}}$ **long-term memory formation** (Conf: 0.75, Rel: 0.80, Clr: 0.80)
- **astrocytes** $\xrightarrow{\text{generates}}$ **extracellular ROS** (Conf: 0.75, Rel: 0.80, Clr: 0.80)
- **extracellular ROS** $\xrightarrow{\text{imported by}}$ **neurons** (Conf: 0.45, Rel: 0.80, Clr: 0.80)
- **Alzheimer's disease model** $\xrightarrow{\text{impairs}}$ **astrocyte-to-neuron H2O2 signaling** (Conf: 0.75, Rel: 0.80, Clr: 0.80)
- **Alzheimer's disease model** $\xrightarrow{\text{impairs}}$ **memory formation** (Conf: 0.85, Rel: 0.80, Clr: 0.80)
- **ROS signaling** $\xrightarrow{\text{important for}}$ **memory** (Conf: 0.75, Rel: 0.70, Clr: 0.60)
- **astrocyte function** $\xrightarrow{\text{important for}}$ **memory** (Conf: 0.75, Rel: 0.80, Clr: 0.60)
- **Alzheimer's disease** $\xrightarrow{\text{involves}}$ **astrocyte function** (Conf: 0.75, Rel: 0.70, Clr: 0.50)
- **Alzheimer's disease** $\xrightarrow{\text{involves}}$ **ROS signaling** (Conf: 0.75, Rel: 0.80, Clr: 0.80)

---

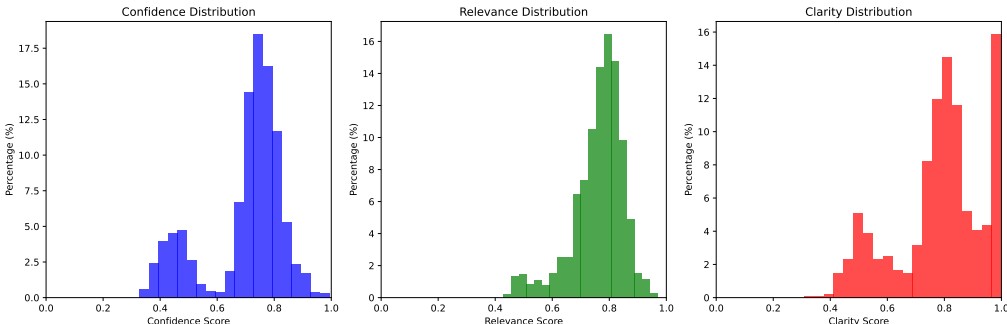

Figure 6: Distribution of confidence, relevance, and clarity scores of extracted metabolomics knowledge graph triples from KARMA.

**Key observations:** High-clarity relationships ($clr \geq 0.8$) typically involve well-characterized biochemical processes, while lower confidence scores often reflect novel or context-dependent findings requiring expert validation.

