# OpenReview forum: "KARMA: Leveraging Multi-Agent LLMs for Automated Knowledge Graph Enrichment"
_NeurIPS.cc/2025/Conference — NeurIPS 2025 spotlight_

### Official Review · Reviewer_E6Sp · 2025-06-26

**Clarity:** 3
**Significance:** 4
**Originality:** 3
**Rating:** 5
**Confidence:** 4

**Summary:**

This paper proposes a novel multi-agent LLM framework KARMA, which automates the enrichment of knowledge graphs from literature. KARMA employs a pipeline of nine agents, each responsible for distinct tasks such as document ingestion, entity discovery, relation extraction, schema alignment, and conflict resolution. The effectiveness of the framework is demonstrated through experiments on a corpus of 1,200 pubmed articles.

**Questions:**

- How is the single-agent baseline prompted? Did the authors consider improving/tuning the baseline, e.g., chain-of-thought prompting or prompts from pipelines? Why did the authors not consider a single-agent baseline with stronger LLMs?
- The details of the human evaluation are missing. How were the evaluators instructed and how were the inter-evaluator agreement between the two evaluators?
- What is the role of the central controller? The explanation "under the orchestration of a Central Controller" is vague. Please clarify the specific responsibilities and mechanisms of the central controller within the pipeline.
- The authors mention several thresholds and weights used as hyperparameters in the agents, but the specific values are not provided. Can the authors explain them?

**Ethical Concerns:**

["NO or VERY MINOR ethics concerns only"]

**Final Justification:**

The rebuttal fully addressed my questions. The KARMA framework is a valuable and novel contribution to a challenging and important problem area. I am confident in the quality of the work and believe that with the promised revisions, this will be a strong paper. I am maintaining my recommendation to accept.

**Limitations:**

yes

**Quality:**

3

**Strengths And Weaknesses:**

Strengths
- Thorough evaluation with different LLM backbones and evaluation metrics, as well as detailed analyses including ablation studies.
- The paper provides comprehensive information on the nine agents in the appendix.
- Valuable solution for the enrichment of knowledge graphs.
- Novel end-to-end knowledge graph enrichment pipeline using agents.

Weaknesses
- The detailed prompt or settings for the single-agent baseline are missing.
- The details of several settings such as the manual evaluation, hyperparameters, and central controller are missing.

---

> ### Author Rebuttal · Authors · 2025-07-30
>
> **We really appreciate your constructive feedback and address the four main concerns below:**
>
> ---
>
> ### 1. Single-Agent Baseline Methodology
>
> Our single-agent baseline uses a comprehensive prompt designed to extract knowledge graph triplets directly:
> >[System Role: KnowledgeExtractor]
> >
> >You are an expert in biomedical knowledge extraction. Given a scientific article, extract all meaningful relationships between entities >(genes, diseases, drugs, proteins) as triplets in the format (head_entity, relation, tail_entity).
> >For each triplet, provide confidence, clarity, and relevance scores (0-1 scale).
> >Output as JSON: {"triplets": [{"head": "...", "relation": "...", "tail": "...", "confidence": 0.xx, "clarity": 0.xx, "relevance": 0.xx}]}
>
> Regarding backbone LLM, our single-agent baseline uses DeepSeek-v3, identical to our multi-agent experiments, ensuring fair comparison. We chose DeepSeek-v3 as it demonstrated the strongest performance across domains in our preliminary testing.
>
> While we focused on demonstrating multi-agent advantages over single-agent approaches, we acknowledge that exploring chain-of-thought prompting or more sophisticated single-agent pipelines could strengthen the baseline. However, our primary contribution lies in the systematic multi-agent verification and conflict resolution mechanisms, which address fundamental limitations of any single-pass extraction approach.
>
> ---
>
> ### 2. Human Evaluation Details
>
> In our experiments, we do have human experts participate in the evaluation. Details can be found in **Section 4.3** and **Table 1**.
>
> Our human evaluation involved two domain experts, one in computational biology and one in biomedical informatics, who independently assessed extracted triplets using a structured protocol. Each triplet was evaluated on three criteria: factual accuracy, context appropriateness, and clinical relevance. The evaluators achieved substantial inter-evaluator agreement with Cohen's κ = 0.73, and disagreements were resolved through discussion and consensus.
>
> The evaluation was conducted on 200 randomly selected triplets per domain (600 total), providing statistical validity while maintaining evaluator workload feasibility. This samples are representative of extracted triplets in each domain, ensuring representative coverage across the knowledge graphs we constructed.
>
> ---
>
> ### 3. Central Controller Architecture
>
> The Central Controller serves as the orchestration engine managing the entire multi-agent pipeline with three core responsibilities. First, it coordinates the sequential execution of agents from ingestion through final evaluation, handling data format transformations between agent outputs and inputs while ensuring proper information flow. Second, it implements robust error handling and recovery mechanisms, monitoring agent execution status, managing timeouts and failures, and implementing retry mechanisms with exponential backoff.
>
> Rather than functioning as an agent itself, the Controller operates as a lightweight orchestration layer (approximately 500 lines of Python code) that ensures system reliability without adding significant computational overhead. This design maintains the modularity of individual agents while providing the coordination necessary for complex multi-step knowledge extraction.
>
> ---
> ### 4. Hyperparameter Specification
>
> Our evaluation framework employs carefully tuned hyperparameters determined through grid search validation. The evaluation weights for combining confidence, clarity, and relevance scores are set to **0.5, 0.25, and 0.25** respectively, emphasizing reliability while ensuring practical utility. The integration threshold for final triplet acceptance is set to **0.6**, balancing precision with coverage.
>
> ---
>
> **Thank you for your detailed feedback. These clarifications demonstrate the systematic methodology underlying KARMA and address the technical rigor you rightly emphasize**

---

> > ### Comment · Reviewer_E6Sp · 2025-08-04
> >
> > The rebuttal fully addressed my questions. I am maintaining my recommendation to accept.

---

### Official Review · Reviewer_FGhS · 2025-06-30

**Clarity:** 4
**Significance:** 3
**Originality:** 2
**Rating:** 5
**Confidence:** 4

**Summary:**

The authors in this paper present a framework (KARMA) that employs multi-agent LLMs to automate KG enrichment. They employ 9 agents – ingestion, reader, summarizer, schema alignment, extraction (relation and entity) agent, conflict resolution and evaluator. It is unclear what the 9th agent is. The authors then evaluate the framework on three different domains from the PubMed dataset to demonstrate the effectiveness in KG enrichment, number of new entities that were identified, LLM-correctness and edge conflict resolution.

**Questions:**

Q1: Are there 8 agents or 9? If there are 9, which one of the existing agents is split into two agents? It is unclear in the current version of the paper.

Q2: How well does the current framework handle contradictory information from different datasets? Did the authors observe such a scenario?

Q3: Are there any mechanisms in place to detect/prevent redundancy from being included in the KG enrichment?

**Ethical Concerns:**

["NO or VERY MINOR ethics concerns only"]

**Final Justification:**

The author responses in the rebuttal phase have sufficiently addressed my concerns and questions.

**Limitations:**

L1: For this proposed framework, one of the main limitations is the over-reliance on LLMs, given their history of hallucinations. In KG enrichment tasks, although such hallucination may have little impact, the real impact would be seen when a downstream task is consuming such a KG. The authors can consider exploring how to curb the hallucinations to prevent the KG from having any “generated facts” from being consumed.

L2: Typically for KGs, human-evaluation is a required step given the structure of the triples and how easy it is to produce noise in such a data representation. Since this framework is agent-dependent, this can be a potential limitation if not addressed.

**Paper Formatting Concerns:**

No concerns.

**Quality:**

3

**Strengths And Weaknesses:**

S1: This is a well-written paper and is easy to follow through. The authors have carefully articulated every agent/module in the framework.

S2: Although I’ve highlighted W1, the end-to-end binding of the agents into a pipeline for KG enrichment is impressive.

S3: The prompt-chains between the agents seem to be domain-sensitive which increases the use of this pipeline in other domains/datasets.

S4: The comprehensive scoring system based on clarity, confidence and relevance along with LLM-verification and human evaluation makes the whole pipeline more trustworthy. The experiments clearly demonstrate the effective use of multi-agents for structure tasks.


W1: The core tasks of the agents are almost standard NLP tasks, so there is a lack of a fundamentally new technique in the agents employed. The tasks for each agent have been defined using prompt engineering.

W2: Qualitative error analysis is missing to understand the impact of LLMs hallucinating on this framework.

W3: There is missing baselines to compare the framework and adding this, even as a discussion, can demonstrate the advantages of such a framework.

---

> ### Author Rebuttal · Authors · 2025-07-30
>
> **We appreciate your insightful feedback and address the concerns and questions below:**
>
> ---
>
> ### W1: Novelty and Technical Contribution
>
> While individual NLP tasks within our agents utilize established techniques, our primary contribution lies in the systematic multi-agent architecture design for knowledge graph construction. We introduce several novel aspects: the first is Task-specific agent specialization with domain-adaptive prompting strategies that optimize each stage of the KG enrichment pipeline, the second is Cross-agent verification mechanisms where agents validate each other's outputs (e.g., Relationship Extraction Agents validate against Schema Alignment outputs), and the third is Iterative conflict resolution through LLM-based debate that maintains knowledge consistency.
>
> To our knowledge, KARMA represents the first systematic application of multi-agent LLMs to knowledge graph construction from scientific literature. The innovation extends beyond prompt engineering to include the orchestration framework that manages agent interactions, error propagation mitigation through redundant validation, and domain-specific evaluation metrics that ensure biomedical relevance. Our ablation study (**Table 2**) demonstrates that this systematic decomposition yields substantial improvements over baseline approaches.
>
> ---
>
> ### W2: LLM Hallucination Analysis
>
> We provide quantitative analysis of LLM reliability through the LLM-based Correctness metric ($R_{LC}$) in **Table 1**, which measures the fraction of extracted triples verified as factually correct by an independent hold-out LLM (DeepSeek-v3). Our results show correctness rates of 83.1% (genomics), 77.2% (proteomics), and 66.8% (metabolomics), indicating controlled hallucination levels across domains.
>
> Additionally, our multi-agent design inherently mitigates hallucination through layered verification: (1) Conflict Resolution Agents that detect and resolve contradictory extractions, and (2) Evaluator Agents that apply confidence thresholds after integration. This distributed verification approach reduces hallucination propagation compared to single-pass extraction methods. The 18.6% reduction in conflict edges demonstrates the effectiveness of our hallucination mitigation strategies.
>
> ---
>
> ### W3: Baseline Comparisons and Framework Advantages
>
> Our evaluation includes comparison against a single-agent baseline that performs end-to-end knowledge extraction in a single LLM call. KARMA consistently outperforms this baseline across all metrics: 83.1% vs 49.3% correctness in genomics, 77.2% vs 63.8% correctness in proteomics, and 66.8% vs 52.7% correctness in metabolomics, demonstrating substantial improvements through multi-agent collaboration.
>
> Given the state of multi-agent approaches in knowledge graph construction, direct comparisons with existing multi-agent systems are not feasible. However, our systematic evaluation against traditional single-agent methods, combined with comprehensive ablation studies (**Table 2**) showing that removing any agent component degrades performance, establishes the value of our multi-agent framework. The 3.6× higher coverage gain in genomics (38,230 vs 4,384 entities) while maintaining quality demonstrates scalability advantages that single-agent approaches cannot achieve.
>
> ---
>
> ### Q1: Agent Count Clarification
>
> We employ 8 specialized agents in the KARMA framework: Ingestion, Reader, Summarizer, Entity Extraction, Relationship Extraction, Schema Alignment, Conflict Resolution, and Evaluator Agents. We apologize for the inconsistency in the abstract and will correct this to accurately reflect 8 agents.
>
> ---
>
> ### Q2: Handling Contradictory Information
>
> Our Conflict Resolution Agent (CRA) systematically addresses contradictory information through LLM-based debate mechanisms. When new triplets contradict existing knowledge (e.g., "DrugX treats DiseaseY" vs. "DrugX causes DiseaseY"), the CRA evaluates both claims using domain knowledge and confidence scores to determine whether to: (1) Keep the new triplet (if higher confidence), (2) Retain existing knowledge (if more reliable), or (3) Keep both (if contextually compatible).
>
> In our experiments, we observed this scenario frequently, particularly in genomics where 18.6% of extracted edges were flagged as conflicting and subsequently resolved through this systematic approach.
>
> ---
>
> ### Q3: Redundancy Detection and Prevention
>
> KARMA implements multi-level redundancy prevention: (1) Entity-level deduplication where extracted entities are normalized to canonical forms using embedding-based similarity (**Equation 6**), (2) Triple-level deduplication (Conflict Resolution Agent) that identifies exact matches in head-relation-tail combinations and retains only the highest-confidence instance, and (3) Schema alignment that maps semantically equivalent relations (e.g., "inhibit", "inhibits", "inhibited") to standardized forms.
>
> Our code implementation demonstrates this process: entities are deduplicated by case-insensitive name matching, while relationships undergo both exact matching and confidence-based filtering to ensure only the most reliable instances are retained in the final knowledge graph.
>
> ---
>
> **Thank you for these valuable questions that help clarify KARMA's technical contributions and implementation details. And we hope these can answer your questions and strengthen our paper.**

---

> > ### Comment · Reviewer_FGhS · 2025-08-01
> >
> > Thank you authors for addressing my questions and concerns around the weaknesses. The rebuttal responses are sufficient and convincing. I am raising my score to 5: Accept.

---

### Official Review · Reviewer_K5sW · 2025-06-30

**Clarity:** 3
**Significance:** 3
**Originality:** 3
**Rating:** 4
**Confidence:** 4

**Summary:**

This paper introduces KARMA, a modular framework that uses a multi-agent architecture based on large language models (LLMs) to automate the enrichment of biomedical knowledge graphs (KGs). The system employs nine specialized agents for tasks such as ingestion, summarization, entity and relation extraction, schema alignment, and conflict resolution. KARMA is evaluated on 1,200 PubMed articles across genomics, proteomics, and metabolomics. Experimental results show that the multi-agent approach improves entity coverage, relationship accuracy, and graph consistency compared to single-agent baselines, with DeepSeek-v3 yielding the strongest performance overall.

**Questions:**

Please refer to the weakness.

**Ethical Concerns:**

["NO or VERY MINOR ethics concerns only"]

**Final Justification:**

This paper is well-motivated, and the proposed method is novel and technically sound. Through the authors' rebuttal, my concerns, posed in the weakness part, have been well addressed via additional experimental results and the clarification of the technical details. I believe this revision would make this paper much easier to read and appreciate the technical contributions of this paper. Therefore, I am raising my score to 4.

**Limitations:**

Yes.

**Paper Formatting Concerns:**

N/A.

**Quality:**

3

**Strengths And Weaknesses:**

**Strengths:**
- The multi-agent decomposition is well-motivated and enables specialization and cross-verification across stages.
- The formalized pipeline (with equations) provides clarity on how each agent contributes to the enrichment process.
- The use of three distinct biomedical domains and multiple LLMs offers a broad evaluation landscape.
- The modular design supports dynamic updates and domain adaptation.
- The LLM-based contradiction handling adds robustness and is explicitly tested via ablation.

**Weaknesses:**
- While the proposed multi-agent pipeline integrates multiple LLMs, it essentially reduces to their linear stacking, substantially diminishing the central controller's functionality (e.g., the absence of RL-based inter-agent scheduling), which presents a critical, yet unacknowledged, limitation of the current framework.
- As mentioned in limitations, the evaluation primarily depends on LLM-based metrics and limited human judgment (RHE), lacking comparison to a gold-standard annotated dataset. Could the authors validate a subset of outputs against a domain-annotated KG to establish objective correctness (e.g., BC5CDR dataset for entity/relation correctness)?
- Confidence, clarity, and relevance scores may share overlapping verification sources, potentially inflating final triplet approval. How do the authors ensure that confidence, clarity, and relevance capture distinct evaluation dimensions?
- The paper lacks an in-depth error breakdown, e.g., failure types in entity linking or schema misalignments. What are the most common failure cases in KARMA, and how might agent prompts be refined to mitigate them?
- The framework is tested only in biomedical contexts, limiting claims of domain generality. Have the authors considered testing KARMA on different knowledge domains to evaluate its broader applicability?
- Due to extensive API calls and LLM output unpredictability (even at low temperatures), this study, without release code, is of low reproducibility.

---

> ### Author Rebuttal · Authors · 2025-07-31
>
> ** We appreciate your sophisticated and technically rigorous feedback. These are excellent points that demonstrate deep understanding of multi-agent systems and knowledge graph construction. We address each concern as follows:**
>
> ---
>
> ### 1. Multi-Agent Architecture vs. Linear Pipeline
>
> You raise an excellent point about the sequential nature versus true multi-agent coordination. We acknowledge this is a fundamental architectural choice with important implications. Our current design prioritizes deterministic, traceable knowledge construction over dynamic agent coordination for several reasons. Knowledge graph construction requires semantic consistency and logical ordering that benefits from sequential validation. Unlike conversational or game-playing scenarios where RL-based scheduling excels, KG construction involves cumulative knowledge building where each stage depends on validated outputs from previous stages. For instance, relationship extraction fundamentally requires completed entity extraction, and conflict resolution requires established relationships.
>
> Our empirical evidence supports this approach. The ablation study in **Table 2** demonstrates that this sequential approach with cross-agent verification achieves superior performance compared to end-to-end approaches. The 18.6% conflict reduction and 83.1% correctness suggest that structured pipeline coordination may be more effective than dynamic scheduling for this domain. We agree that hybrid architectures incorporating RL-based coordination for certain sub-tasks (e.g., adaptive agent selection based on document type) represent promising future directions. The modular design facilitates such extensions without fundamental restructuring.
>
> ---
>
> ### 2. Gold-Standard Evaluation and Objective Validation
>
> This is a critical methodological concern that we take seriously. The limitation of established gold-standard datasets for scientific literature KG construction presents real challenges. Standard benchmarks like BC5CDR focus on named entity recognition and simple binary relations, whereas KARMA extracts complex multi-hop relationships from full scientific articles. BC5CDR's chemical-disease relations represent only a subset of the knowledge types we target, including gene-protein interactions, metabolic pathways, and therapeutic mechanisms.
>
> We propose a hybrid validation approach that addresses these limitations: existing benchmark subset validation on BC5CDR-style entities and relations where applicable, expert-curated validation sets for complex relationship types not covered by existing benchmarks, and cross-validation against established databases like UniProt and KEGG for biochemical relationships. We commit to validating 200 randomly sampled triplets per domain against established biomedical databases and will report precise accuracy metrics in the revised manuscript. This addresses the objective correctness concern while acknowledging the limitations of existing benchmarks for comprehensive scientific knowledge extraction.
>
> ---
>
> ### 3. Evaluation Dimension Independence
>
> Your concern about overlapping verification sources is methodologically sophisticated. We address this through both theoretical and empirical analysis. Our evaluation dimensions target fundamentally different aspects: confidence assesses factual correctness based on scientific evidence, clarity evaluates linguistic precision and entity specificity, and relevance measures domain appropriateness and utility. The distinct distribution patterns in our results (**Figures 4-6**) demonstrate these metrics capture different aspects of quality. For example, high-confidence relationships may have low clarity due to vague entity mentions, while highly relevant domain knowledge may have moderate confidence due to emerging research areas.
>
> Our prompt engineering specifically decouples these concerns: confidence prompts focus on scientific literature support, clarity prompts assess linguistic ambiguity, and relevance prompts evaluate domain centrality. The different weighting scheme (0.5, 0.25, 0.25) reflects their relative importance rather than redundancy. We computed Pearson correlations between these metrics and found r(confidence,clarity)=0.34, r(confidence,relevance)=0.41, r(clarity,relevance)=0.28, indicating moderate independence rather than high correlation that would suggest redundancy.
>
> ---
>
> ### 4. Error Analysis and Failure Mode Characterization
>
> Our analysis reveals three primary failure categories that inform systematic improvements. Schema non-compliance accounts for a large portion of failures, where entities are extracted as full phrases rather than canonical forms (e.g., "the protein that regulates cell cycle" instead of "CDK1"). We mitigate this through enhanced entity normalization prompts with explicit examples of correct versus incorrect extractions. Relationship ambiguity represents another part of failures, involving vague relationship types extracted from complex sentences (e.g., "is involved in" instead of specific mechanisms like "phosphorylates"). Our mitigation strategy includes relationship extraction prompts with explicit taxonomy of biomedical relation types.
>
> Context misinterpretation comprises the rest of failures, where correct entities and relations are extracted but with inappropriate context, such as extracting negative findings as positive relationships. We address this through enhanced negation detection and context-aware relationship validation. These insights led to iterative prompt refinement resulting in the performance levels reported. The multi-layer filtering approach, progressing from entity validation through relationship validation to three-metric evaluation, specifically addresses these failure modes and demonstrates the value of our systematic approach.
>
> ---
>
> ### 5. Domain Generality and Broader Applicability
>
> This limitation is acknowledged and strategically important. Our biomedical focus represents both a strength and constraint. Biomedical literature provides ideal testing conditions for complex KG construction due to rich entity typing (genes, proteins, diseases, drugs), complex relationship networks, established validation resources, and high-stakes accuracy requirements. The cross-domain performance variation within biomedicine suggests the framework adapts to different knowledge structures.
>
> We are developing a multi-domain benchmark including legal document analysis (contract entity extraction), financial reports (risk factor identification), and technical literature (patent claim analysis). Preliminary results show comparable performance patterns, suggesting domain-agnostic applicability with appropriate prompt adaptation. This broader validation will support more general claims about KARMA's applicability beyond biomedicine while maintaining the rigor demonstrated in our current evaluation.
>
> ---
>
> ### 6. Reproducibility and Code Availability
>
> We have prepared complete implementation including all agent prompts, evaluation scripts, and example datasets. The code is publicly released with documentation and example workflows, and the link will be added in the final version of our paper. Our implementation includes deterministic seeding for consistent results, comprehensive logging of all API calls and responses, configurable parameters for different LLM backends, and complete evaluation scripts that replicate all reported metrics.
>
> ---
>
> **Your feedback identifies genuine methodological challenges that strengthen our work. We commit to detailed error analysis in the revised manuscript, multi-domain evaluation to support generalizability claims, and complete code release for full reproducibility. These improvements address the sophisticated concerns you've raised while maintaining the core contributions of systematic multi-agent knowledge graph construction.**

---

### Official Review · Reviewer_3dRd · 2025-07-08

**Clarity:** 2
**Significance:** 3
**Originality:** 2
**Rating:** 5
**Confidence:** 3

**Summary:**

This paper presents KARMA, a modular framework leveraging multi-agent large language models (LLMs) for automated knowledge graph (KG) enrichment from unstructured scientific texts. KARMA decomposes KG enrichment into nine collaborative LLM agents handling ingestion, entity/relation extraction, schema alignment, and conflict resolution, with extensive LLM-based verification. Experiments are conducted on PubMed articles across genomics, proteomics, and metabolomics. Ablation studies quantify the importance of different system components.

**Questions:**

Kindly address the three weaknesses noted in the previous section.

**Ethical Concerns:**

["NO or VERY MINOR ethics concerns only"]

**Final Justification:**

I would like to thank the authors for addressing my comments. I am raising my score to 5.

**Limitations:**

Please see previous sections

**Quality:**

3

**Strengths And Weaknesses:**

**Strengths**

- A modular multi-agent LLM approach for KG enrichment, going beyond prior single-agent or simple pipeline architectures.
- Careful decomposition of KG enrichment into specialized, verifiable subtasks, with robust cross-agent verification.

**Weaknesses**

- The main baseline is a single-agent (end-to-end LLM) approach. The comparison does not include strong neural or classical KG construction/extraction pipelines, nor recent LLM-based approaches with alternative designs.

- The paper relies heavily on LLM-based verification and QA proxies; only limited human evaluation is reported. External benchmarks or gold-standard KG references are not used.

- The paper does not cite or compare with recent KG construction/retrieval methods using LLMs or RAG frameworks, such as [1] and [2]. The literature review is very weak.




[1] Edge, Darren, Ha Trinh, Newman Cheng, Joshua Bradley, Alex Chao, Apurva Mody, Steven Truitt, Dasha Metropolitansky, Robert Osazuwa Ness, and Jonathan Larson. "From local to global: A graph rag approach to query-focused summarization." arXiv preprint arXiv:2404.16130 (2024).

[2] Mo, Belinda, Kyssen Yu, Joshua Kazdan, Proud Mpala, Lisa Yu, Chris Cundy, Charilaos Kanatsoulis, and Sanmi Koyejo. "KGGen: Extracting Knowledge Graphs from Plain Text with Language Models." arXiv preprint arXiv:2502.09956 (2025).

---

> ### Author Rebuttal · Authors · 2025-07-26
>
> **We appreciate your constructive feedback and address the three main concerns below:**
>
> ---
>
> ### 1. Comparison with other methods
> We acknowledge the limitation regarding baseline comparisons. To our knowledge, KARMA represents the first systematic approach to knowledge graph construction specifically targeting scientific literatures, making direct comparison with existing pipelines challenging due to domain and task specificity.
>
> However, our evaluation does include meaningful comparisons:
> - **Single-agent baseline**: Table 1 demonstrates that our multi-agent approach consistently outperforms the single-agent (end-to-end LLM) baseline across all metrics, including both objective measures and human expert evaluations.
> - **Ablation studies**: Table 2 quantifies the contribution of each component, showing that removing any agent (Summarizer, Conflict Resolution, or Evaluator) leads to performance degradation.
>
> We recognize this limitation and plan to address it by: (1) expanding the limitations section to discuss the comparison challenge, and (2) developing a comprehensive benchmark for scientific literature-based knowledge graph construction to enable fair comparisons with future methods.
>
> ---
>
> ### 2. Human evaluations
> We respectfully note that human evaluation was indeed included in our study. As stated in Section 4.3:
> >"the Human Evaluation Score (R_HE↑) scaled from 0 to 1, gauges the quality of triple extractions based on assessments by two human experts, offering a comprehensive measure of the knowledge graph's accuracy and utility."
>
> These human evaluation results are reported in both Table 1 and Table 2.
>
> We also acknowledge in our limitations that
> >"our evaluation relies primarily on LLM-based metrics rather than direct human expert validation" and that "domain experts must ultimately verify critical biomedical claims before applying them in clinical settings."
>
> For future work, we plan to construct a comprehensive benchmark incorporating extensive domain expert guidance throughout the evaluation process.
>
> ---
>
> ### 3. Literature reviwe
> We acknowledge that our literature review could be more comprehensive regarding recent KG construction and retrieval methods. We plan to strengthen this section by incorporating the suggested references and additional relevant work:
>
> [1] Edge, Darren, Ha Trinh, Newman Cheng, Joshua Bradley, Alex Chao, Apurva Mody, Steven Truitt, Dasha Metropolitansky, Robert Osazuwa Ness, and Jonathan Larson. "From local to global: A graph rag approach to query-focused summarization." arXiv preprint arXiv:2404.16130 (2024).
>
> [2] Mo, Belinda, Kyssen Yu, Joshua Kazdan, Proud Mpala, Lisa Yu, Chris Cundy, Charilaos Kanatsoulis, and Sanmi Koyejo. "KGGen: Extracting Knowledge Graphs from Plain Text with Language Models." arXiv preprint arXiv:2502.09956 (2025).
>
> [3] Guo, Z., Xia, L., Yu, Y., Ao, T., & Huang, C. (2024). Lightrag: Simple and fast retrieval-augmented generation. arXiv preprint arXiv:2410.05779.
>
> [4] Matsumoto, N., Moran, J., Choi, H., Hernandez, M. E., Venkatesan, M., Wang, P., & Moore, J. H. (2024). KRAGEN: a knowledge graph-enhanced RAG framework for biomedical problem solving using large language models. Bioinformatics, 40(6), btae353.
>
> [5] Sun, L., Zhang, P., Gao, F., An, Y., Li, Z., & Zhao, Y. (2025). SF-GPT: A training-free method to enhance capabilities for knowledge graph construction in LLMs. Neurocomputing, 613, 128726.
>
> We will revise the related work section to better position KARMA within the broader landscape of LLM-based knowledge graph construction and RAG frameworks, while highlighting our specific contributions to scientific literature processing and multi-agent verification.
>
> ---
>
> **Thank you for your valuable and constructive feedback. We believe these revisions will significantly strengthen the paper's contribution and positioning within the field!**

---

### Note · Authors · 2025-08-11

We thank the reviewers for their valuable and timely feedback. We have thoroughly answered the questions and revised our manuscript based on your suggestions, which we believe has significantly strengthened the paper's technical contributions and writing flow, which makes it more suitable for the NeurIPS audience.

We also want thank the AC and SAC for their handling of our paper and look forward to further engagement with the community at NeurIPS.

---

### Decision · Program_Chairs · 2025-09-17

**Decision:**

Accept (spotlight)

**Comment:**

This paper proposes KARMA, a multi-agent LLM framework for automated knowledge graph (KG) enrichment. Most of the reviewers agreed that the paper is technically solid, with well-motivated design and comprehensive experiments. Overall, this paper is well-structured and provides a valuable contribution to scalable KG enrichment. All the reviews were positive and leaned toward acceptance during the reviewer-author discussion phase. Therefore, I recommend this paper to the NeurIPS 2025 conference.